# Outlier response to anti-PD1 in uveal melanoma reveals germline *MBD4* mutations in hypermutated tumors

Manuel Rodrigues [1,2], Lenha Mobuchon[1], Alexandre Houy[1], Alice Fiévet[1,3], Sophie Gardrat[4], Raymond L. Barnhill[4,5], Tatiana Popova[1], Vincent Servois[6], Aurore Rampanou[7], Aurore Mouton[8], Stéphane Dayot[1], Virginie Raynal[9], Michèle Galut[10], Marc Putterman [11], Sarah Tick[12], Nathalie Cassoux[5,13], Sergio Roman-Roman[14], François-Clément Bidard[2,7,15], Olivier Lantz [8], Pascale Mariani[16], Sophie Piperno-Neumann[2] & Marc-Henri Stern [1,3]

Metastatic uveal melanoma is a deadly disease with no proven standard of care. Here we present a metastatic uveal melanoma patient with an exceptional high sensitivity to a PD-1 inhibitor associated with outlier CpG>TpG mutation burden, *MBD4* germline deleterious mutation, and somatic *MBD4* inactivation in the tumor. We identify additional tumors in The Cancer Genome Atlas (TCGA) cohorts with similar hypermutator profiles in patients carrying germline deleterious *MBD4* mutations and somatic loss of heterozygosity. This *MBD4*-related hypermutator phenotype may explain unexpected responses to immune checkpoint inhibitors.

[1] Institut Curie, PSL Research University, INSERM U830, DNA Repair and Uveal Melanoma (D.R.U.M.), Equipe labellisée par la Ligue Nationale contre le Cancer, Paris 75248, France. [2] Department of Medical Oncology, Institut Curie, PSL Research University, Paris 75248, France. [3] Department of Genetics, Institut Curie, PSL Research University, Paris 75248, France. [4] Department of Biopathology, Institut Curie, PSL Research University, Paris 75248, France. [5] Faculty of Medicine, University of Paris Descartes, Paris 75006, France. [6] Department of Medical Imaging, Institut Curie, PSL Research University, Paris 75248, France. [7] Institut Curie, PSL Research University, Laboratory of Circulating Tumor Biomarkers, SiRIC, Paris 75248, France. [8] Institut Curie, PSL Research University, INSERM CIC-BT 1428, Paris 75248, France. [9] Institut Curie, PSL Research University, INSERM U830 and Institut Curie Genomics of Excellence (ICGex) Platform, Paris 75248, France. [10] Institut Curie, PSL Research University, Biological Resource Center, Paris 75248, France. [11] Department of Pathology, Quinze-Vingts National Ophthalmology Hospital, Paris 75012, France. [12] Department of Ophthalmology III, Quinze-Vingts National Ophthalmology Hospital, Paris 75012, France. [13] Department of Ocular Oncology, Institut Curie, PSL Research University, Paris 75248, France. [14] Department of Translational Research, Institut Curie, PSL Research University, Paris 75248, France. [15] UVSQ, Paris Saclay University, Saint-Quentin 78035, France. [16] Department of Surgical Oncology, Institut Curie, PSL Research University, Paris 75248, France. These authors contributed equally: Olivier Lantz, Pascale Mariani, Sophie Piperno-Neumann. Correspondence and requests for materials should be addressed to M.-H.S. (email: marc-henri.stern@curie.fr)

Uveal melanoma (UM) is an ocular neoplasia most often affecting populations of European ancestry and has one of the lowest mutation burdens among adult tumors[1–3]. Inactivation of *BAP1* (3p21), through both deleterious mutations and monosomy 3, is frequent in UM and is associated with a high risk of metastasis[4]. Prognosis of metastatic UM is dismal with median survival <12 months and no systemic treatment improving survival[3]. Programmed cell death protein 1 inhibitors (PD1inh), a class of immune checkpoint inhibitors, have been evaluated in UM with low overall response rates[5–8]. Here we present three patients with hypermutated CpG>TpG tumors (two UM and one glioblastoma) associated with *MBD4* germline deleterious mutations and somatic inactivation in the tumors. Furthermore, we provide evidence for sensitivity to immune checkpoint inhibitors in *MBD4*-deficient tumors.

## Results

**An outlier metastatic UM patient responding to pembrolizumab.** In our series of 42 metastatic UM patients treated with PD1inh, only one (UVM_IC) achieved a tumor response (details in Methods)[9]. UVM_IC developed a metastatic UM with liver, lung, and bone lesions (Fig. 1a, c and Supplementary Fig. 1). Treatment with the PD1inh pembrolizumab resulted in complete response of known metastases 10 months later, while new non-life-threatening infracentimetric subcutaneous metastases appeared. Longitudinal monitoring of the *GNAQ*^Q209L mutation in circulating tumor DNA was consistent with imaging (Fig. 1b)[10]. We observed peri- and intra-tumoral CD3^+ lymphocytic infiltrates in all UVM_IC samples (Supplementary Fig. 2). After 2 months of pembrolizumab, proportions of blood effector memory CCR7^−/CD45RA^−/CD4^+ and CCR7^−/CD45RA^+/CD8^+ T-cells increased from 14.5% to 21.8% and from 12.2% to

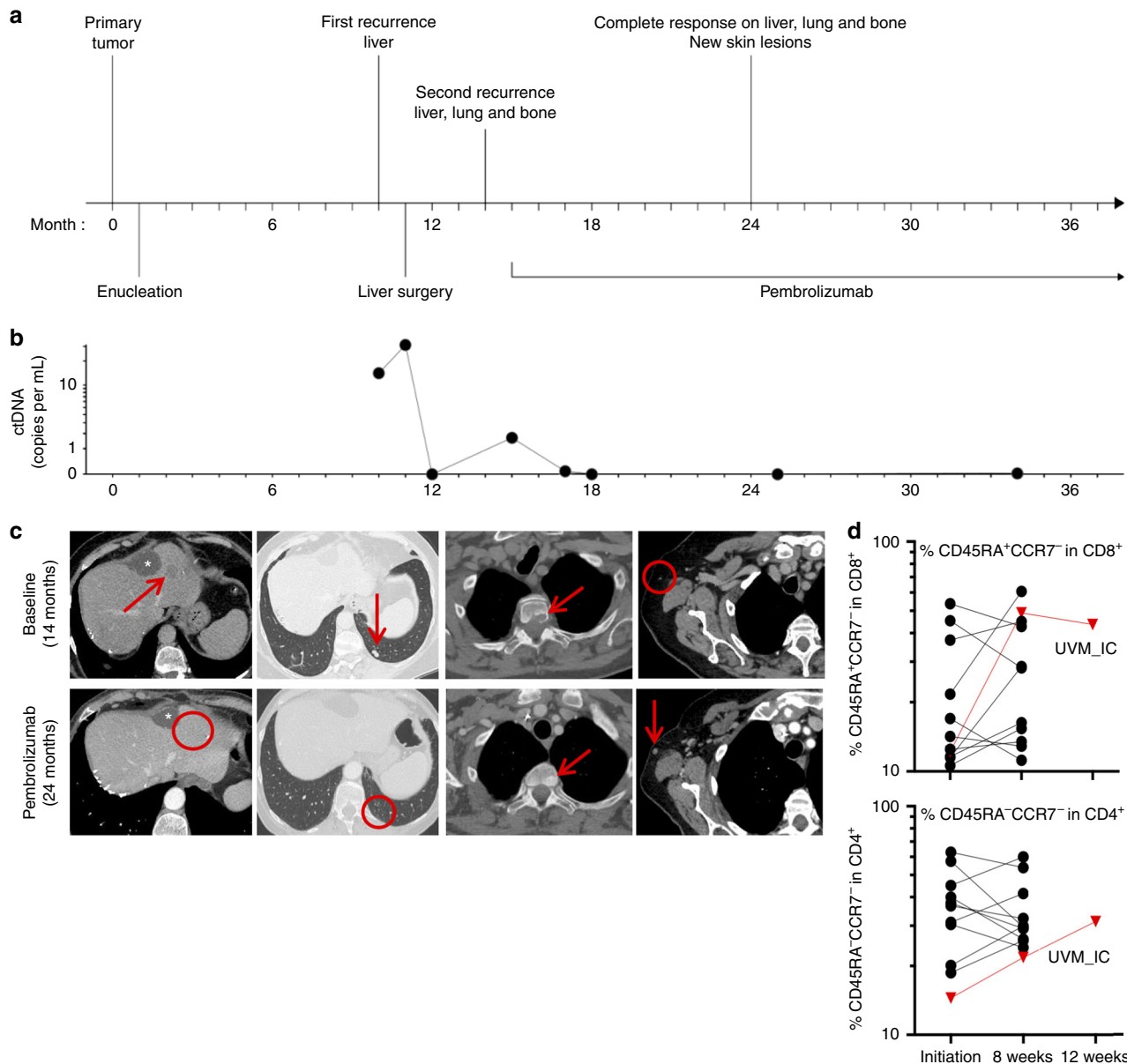

**Fig. 1** Disease course and immune response in patient UVM_IC. **a** Disease course since diagnosis. **b** Evolution of the *GNAQ*^Q209L mutation in circulating tumor DNA (ctDNA). **c** Computed tomography images at second relapse (14 months) and after 10 months of pembrolizumab (24 months). Arrows and circles show locations of metastases; asterisks indicate a simple hepatic cyst. **d** Proportions of blood effector memory CCR7^−/CD45RA^−/CD4^+ and CCR7^−/CD45RA^+/CD8^+ T-cells in 12 metastatic uveal melanoma patients treated with PD-1 inhibitors, including UVM_IC (red line)

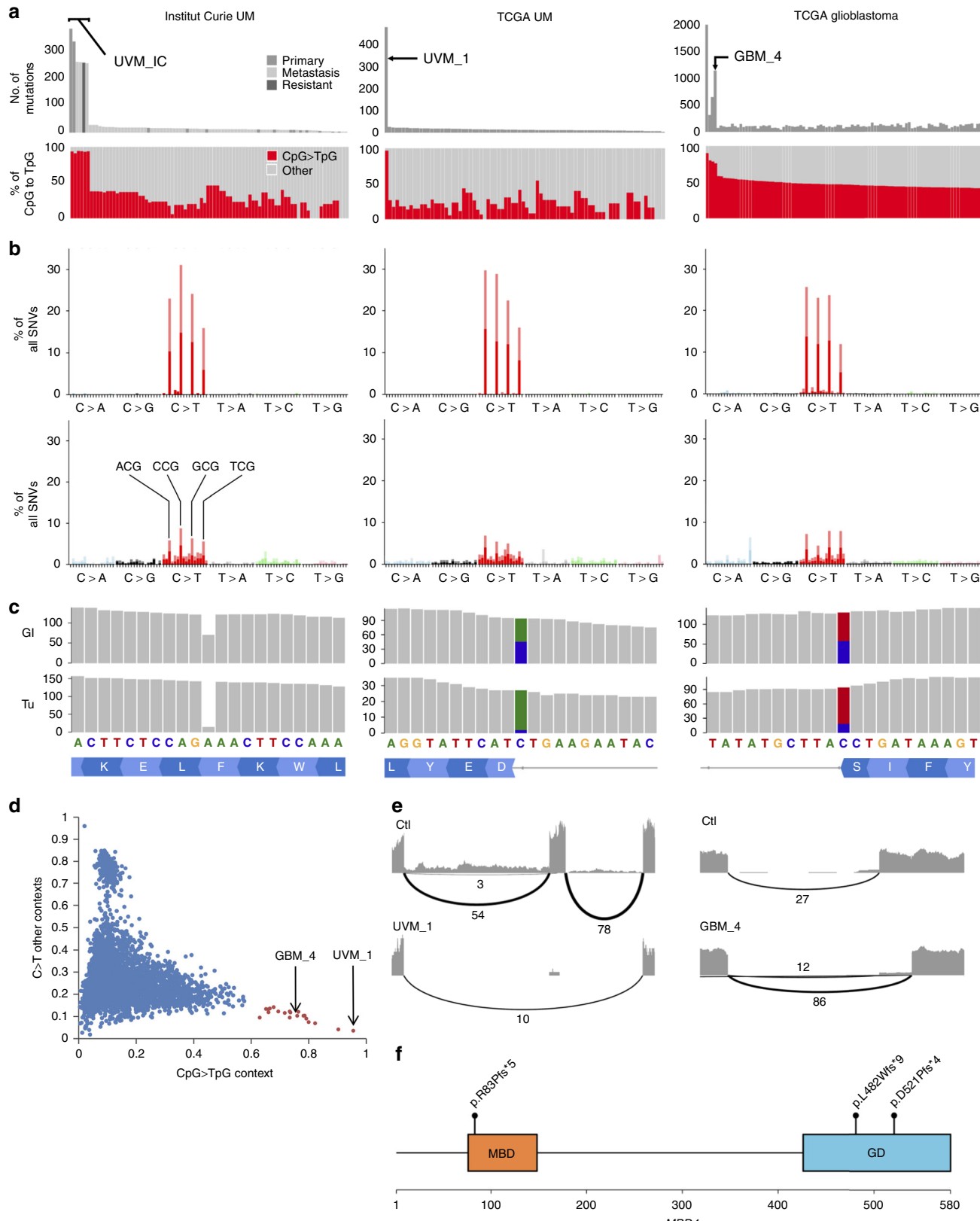

48.9%, respectively. In comparison, the proportions of these populations in 11 non-responsive metastatic UM patients changed modestly (from 38% to 35.3%, and from 23.5% to 27.5%, respectively; Fig. 1d). These observations suggested that pembrolizumab stimulated a previously existing spontaneous cell-mediated immunity against the UVM_IC tumor.

**MBD4 defect is associated with hypermutated CpG>TpG pattern.** To explore this outlier response, we performed whole-exome sequencing (WES) of the primary tumor, the liver metastasis, and a pembrolizumab-resistant subcutaneous metastasis, as well as constitutional DNA. All cancer samples carried somatic $GNAQ^{Q209L}$ and $BAP1^{R385X}$ mutations as well as

**Fig. 2** *MBD4* germline mutations in hypermutated tumors. **a** Number of mutations in tumors from three series: Institut Curie-UM (uveal melanoma; 14 primary and 71 metastatic samples from 23 individuals), TCGA-UM (*n* = 80), and TCGA glioblastoma (*n* = 496; only the 100 with highest proportions of CpG>TpG are shown, full series in Supplementary Fig. 5). Proportion of CpG>TpG mutations versus all other mutations are shown below. **b** Mutational patterns in the tumors of interest (above; from left to right: UVM_IC, UVM_1, and GBM_4) and in the rest of the corresponding series (below). *X*-axis and *Y*-axis indicate the 96 trinucleotide substitutions and the relative proportion of each substitution. Dark and light colors indicate sense and anti-sense strands, respectively. **c** *MBD4* mutations in germline (GI) and tumor (Tu) in the tumors of interest (from left to right: UVM_IC, UVM_1 and GBM_4). **d** TCGA tumors with >200 SNVs are plotted according to proportions of C>T in a CpG context (*X*-axis) and C>T in other contexts (*Y*-axis). The 20 tumors with highest CpG>TpG proportions appear in red. **e** Sashimi plots from RNA-sequencing data. Cases are compared to control (Ctl) tumors from the same series. Only junctions with more than two reads are shown. **f** Location of *MBD4* mutations. GD glycosylase domain, MBD methyl-CpG binding domain

monosomy 3. We identified similar hypermutated profiles in all samples with >266 somatic single nucleotide variants (SNVs) per sample (>176 non-synonymous SNVs) corresponding to >19-fold increase of SNVs compared to an in-house series (Fig. 2a). Over 91% of mutations were CpG>TpG transitions, compared to <30% in other UMs (Fig. 2b). Because CpG>TpG transitions are secondary to the spontaneous deamination of 5-methylcytosines, we searched for alterations of either *TDG* or *MBD4*, encoding two glycosylases involved in 5-methylcytosine integrity[11–13]. We identified a germline deleterious frameshift deletion of *MBD4* (3q21.3; c.1441delT:p.F481Dfs*9) with loss of the second allele by monosomy 3 in all tumor samples (Fig. 2c, f). No other sample in our UM series carried a *MBD4* or *TDG* mutation.

We inferred the clonal structure and observed that the primary tumor presented multiple subclones, which is unusual in UM, while metastases were more homogeneous (Supplementary Figs. 3a and 4). We then observed that each metastasis shared more SNVs with the primary tumor than with other metastases, suggesting polyphyletic clones (Supplementary Fig. 3b, c). Furthermore, each metastasis presented 18–44 new SNVs, again dominated by CpG>TpG (>93%), compared to the predicted initial clones, while cohort analyses demonstrate that UMs usually acquire a mean of two SNVs during metastatic progression (Supplementary Fig. 4). Altogether, these data suggest an ongoing MBD4-related mutagenic process during tumor progression, as has been observed with APOBEC in other cancers[14].

***MBD4* germline mutations in UM and glioblastoma**. To investigate the frequency of *MBD4* and hypermutation in an independent UM cohort, we analyzed the TCGA UM dataset (*N* = 80 patients). This identified an additional hypermutated UM case (patient UVM_1; Fig. 2a).This tumor was carrying a *BAP1* mutation and monosomy 3 as well as 474 SNVs (305 non-synonymous SNVs) corresponding to a 36-fold increase of SNVs as compared to the overall TCGA UM series. Again, the SNVs were predominantly CpG>TpG (460/474; 97% of SNVs). This patient furthermore carried a germline c.1562-1G>T:p.D521Pfs*4 *MBD4* splice-site variant and somatic loss of the wild-type allele due to tumor monosomy 3. Analysis of RNA-seq demonstrated that this splice-site variant was associated with exon 7 skipping and a frameshift (Fig. 2c, e, f). No other *MBD4* or *TDG* mutation was identified in this series. We further analyzed the pan-cancer TCGA series (>10,000 tumors; Supplementary Table 1) and identified 4831 hypermutated tumors (>200 SNVs per tumor) of which 20 cases, including UVM_1, were enriched in CpG>TpG mutations ($\frac{CpG>TpG}{SNVs} > 0.6$; Fig. 2d). Of these 20 cases, patient GBM_4 presented a glioblastoma carrying 1149 SNVs (440 non-synonymous SNVs) and a germline c.335+1G>A:p.R83Pfs*5 *MBD4* mutation with somatic loss of heterozygosity leading to the use of a cryptic splice donor site, loss of 88 bases, and a premature stop codon (Fig. 2c, e, f). The three other hypermutated CpG>TpG glioblastoma cases did not carry any identifiable deleterious *MBD4* or *TDG* mutation. The germline *MBD4*

mutations identified in patients UVM_IC, UVM_1, and GBM_4 are rare in the general population with minor allele frequencies ranging from ~0.000008 to ~0.00002. To be noticed, three of these 20 hypermutated cases carried somatic *MBD4* indels together with mismatch repair deficiency (two colorectal and one endometrial adenocarcinomas); the molecular mechanism of hypermutation in the other cases remains undetermined.

## Discussion

A role for *MBD4* germline mutations in cancer predisposition was hypothesized 18 years ago[13]. The identification of two UM cases with *MBD4* germline mutations is intriguing, and possibly related to the frequent monosomy 3—where *MBD4* is located—in this disease. Integrating our institutional cohort and the TCGA UM cohort, *MBD4* germline deleterious mutations were present in 2% of UM patients (2/102). Both UVM_1 and GBM_4 tumors presented before the age of 50, earlier than median ages (60 in UM and 65 in glioblastoma)[15, 16]. However, none of the three patients had a reported personal or familial history of invasive cancer. In this regard, *Mbd4* knock-out mice models are associated with increase of CpG>TpG transitions without increased tumor incidence, except in *Apc*-deficient backgrounds[17, 18]. Thus, *MBD4* inactivation may not be sufficient to initiate tumorigenesis but may play a significant role in tumor progression.

Because high mutation burden is predictive of response to immune checkpoint inhibitors[19], PD1inh have shown a high activity in hypermutated mismatch repair-deficient tumors leading to the tissue-agnostic approval of PD1inh in these tumors[20]. Hence, while the limited activity of PD1inh in UM patients may be explained by a low mutation burden, the *MBD4*-related high mutation load probably contributed to the dramatic response in the UVM_IC patient. These observations open avenues for clinical trials providing tissue-agnostic access to PD1inh to treat patients with *MBD4*-deficient tumors.

## Methods

**Case reports**. A 76-year-old woman was diagnosed with a stage IIIA UM localized on the left eye choroid (UVM_IC). She had a past medical history of breast ductal carcinoma in situ diagnosed at age 74, and no familial history of cancer. Importantly, no prior mutagen exposure was identified. She underwent enucleation for her UM. Histopathological examination revealed a tumor measuring 10.6 mm in diameter and 10 mm in thickness of mixed spindle cell/epithelioid cell morphology. No scleral or optic nerve infiltration was observed. As her tumor presented with monosomy 3, the patient was deemed at high risk for metastatic disease and close surveillance was instituted. Nine months after enucleation, computed tomography detected a unique liver lesion. Resection of the lesion with clear margins revealed a UM metastasis. Four months later, the patient experienced a relapse with new liver, lung, and osteolytic bone lesions. Treatment with pembrolizumab (2 mg per kilogram of body weight every 3 weeks) was initiated. Ten months later, liver and lung metastases exhibited a complete response, while osteolytic bone lesions showed sclerotic features suggestive of a response. Longitudinal monitoring of circulating tumor DNA (*GNAQ*^Q209L mutation in plasma) was consistent with the clinical and imaging status (Fig. 1b)[10]. The patient has now been receiving pembrolizumab for >22 months without visceral tumor progression. In order to identify the mechanisms implicated in sensitivity to pembrolizumab, immune response markers were analyzed in the primary tumor and in the removed liver metastasis collected prior to pembrolizumab therapy. Pathological examinations showed CD3+ peri- and intra-tumoral lymphocytic infiltrates in both samples (Fig. 1b); however, CD8+ infiltrates were only observed in the primary tumor.

With respect to PD-L1 expression, 30% of the primary tumor was positive for PD-L1 versus 20% in the metastasis. However, 90% of peri-tumoral stroma and immune cells (ICs) of the metastasis expressed PD-L1.UVM_1 is a 41-year-old woman without past medical history of cancer, diagnosed with a stage IIIA UM localized to the choroid. No metastatic recurrence was reported after a follow-up of 39 months. The tumor showed focal lymphocytic infiltration with low CD8A and moderate CD274 (which codes for PD-L1) RNA expression.

GBM_4 presented a glioblastoma at age 48. No other details are available.

**Sample collection**. Several non-interventional cohorts have been proposed in Institut Curie to metastatic UM patients. In ctDNA R0 (ClinicalTrials.gov, NCT02849145) patients provided written informed consent to perform ctDNA monitoring, germline, and somatic genetic analyses of resected metastases/archived primary tumors. In ALCINA (NCT02866149), patients provided written informed consent to perform blood-borne biological markers and correlation with clinical/pathological characteristics. ctDNA R0 and ALCINA were approved by the Internal Review Board of the Institut Curie. UVM_IC was in both cohorts after consenting. Furthermore, we collected and analyzed samples from patients eligible to therapeutic liver R0 resection of UM metastases in our institution including UVM_IC. All patients provided written informed consent to perform germline and somatic genetic analyses of resected metastases and archived frozen primary tumors.

**DNA sequencing**. Samples were histologically reviewed by a pathologist before nucleic acids extraction in order to select samples with at least 30% of melanoma cells. DNAs were extracted from snapped frozen samples, except for the primary tumor sample from UVM_IC, which was extracted from a formalin-fixed paraffin-embedded (FFPE) sample. Germline DNA was extracted from unaffected tissues (healthy liver or blood). DNAs were extracted from frozen samples using phenol (Invitrogen, Carlsbasd, CA, USA) by the Centre de Ressources Biologiques (Institut Curie tumor biobank) and from FFPE using the Nucleospin Tissue kit (Macherey-Nagel GmbH & Co. KG, Düren, Germany) then subsequently purified on Zymo-Spin™ IC (Zymo Research, Irvine, CA, USA). DNAs were quantified by Qubit (Thermo Fisher Scientific, Waltham, MA, USA) and integrity was assessed by BioAnalyzer 2100 (Agilent Technologies, Santa Clara, CA, USA).

WES libraries were prepared using the Agilent SureSelect XT2 Clinical Research Exome kit (Agilent Technologies) from 1 µg of DNA isolated from initial libraries with median size of 300 bp according to the manufacturers' protocols. Libraries were 100 bp paired-end multiplex sequenced on the Illumina HiSeq 2000 (Illumina). WES depth was a priori settled up to sequence germline DNA at 30× and somatic DNA at 100×. The library from UVM_IC primary tumor was prepared using the Agilent SureSelectXT HS kit (Agilent Technologies) from 20 ng of DNA isolated initial libraries with median insert size of 300 bp according to the manufacturers' protocols. Libraries were 100 bp paired-end multiplex sequenced on the Illumina HiSeq 2000 (Illumina). WES depth was a priori settled up at 100×.

**Mutation calling**. Sequencing quality was assessed by FastQC. WES reads were aligned to the human genome (hg19) with Bowtie2 2.1.0[21]. PCR duplicates were removed using Picard Tool MarkDuplicates v1.97. WES data underwent variant calling for SNP and indels using the combination of three variant callers: HaplotypeCaller, MuTect2, and SAMtools mpileup[22–24]. Union of variants detected with these three algorithms were annotated using ANNOVAR[25] with the following databases: ensGene, avsnp147[26], cosmic80[27], popfreq_all_20150413, and dbnsfp33a. Somatic variants with <10 reads of position depth (DP) in germline and/or <10 reads of somatic DP and/or <6 reads of allele depth (AD) and/or an AD/DP ratio of <0.05 and/or a population frequency higher than 1% (popfreq_all >0.01) were filtered out. All somatic mutations called by this procedure were controlled manually using the Integrative Genomics Viewer (IGV).

Sashimi plots were generated with the R software version 3.4.2 and its packages GenomicAlignments (1.12.2) and GenomicRanges (1.28.4) from data downloaded through the GDC portal. Expression data of the tumors from UVM_1 and GBM_4 was explored through the cbioportal. Effect of splice mutations was predicted on the major transcript with the Alamut Visual Software (Interactive Software, Rouen, France), which contains the algorithms SpliceSiteFinder, MaxEntScan, NNSPLICE, GeneSplicer, Human Splicing Finder, ESE-Finder, and RESCUE-ESE. Mutation data from all TCGA series were retrieved from the GDC portal in day 29th of September 2017. Frequencies of germline mutations in general population were retrieved from the ExAC database (access on 12 October 2017).

**Clonal evolution**. Clonal evolution was inferred with Pyclone[28]. Parameters used to define clusters of SNVs were: (i) at least five SNVs and (ii) mean cancer cell fraction of at least 10% in one sample.

**Blood cytometry**. The whole blood was washed in PBS 1X with 0.4 g/l human albumin. Samples were stained with master mix of antibodies for analysis on CantoII or LSRFortessa flow cytometers (BD) before lysis of red cells and fixation (BD FacsTM Lysing solution 10X). Data were analyzed using Flow Jo software (Tree Star).The anti-human antibodies used were CD8b PC5 (2ST8.5H7; Beckman Coulter), TCR γδ FITC (11F2), (BD Biosciences), CD45RA PC7 (HI100 eBiosciences), CD4 PE TX (S3.5 Invitrogen), CD3 Alexa700 (UCHT1), CCR7 BV421

(G043H7), CD27 BV605 (O323), and CD127 BV650 (A019D5)—all from Biolegend.

**Immunohistochemistry**. Immunohistochemistry (IHC) was performed on FFPE sections (4 µm in thickness) for CD3 (Dako, A0452, 1/200), CD4 (Dako 4B12 IR649, 1/100), CD8 (Dako C8/144B, IS623, undiluted), PD-L1 (Dako clone 22C3), MLH1 (Dako ES05 1/50), MSH2 (Dako FE11 1/50), MSH6 (Dako EP49 1/50), PMS2 (BD Phamingen A16-4 1/50). PD-L1 immunostaining with Dako clone 22C3 was performed by Merck Research Laboratory, Palo Alto, CA. IC (lymphocytic) infiltrates were scored according to the method of Rothermel et al.[29]modified to include both (1) percentage of tumor area occupied by tumor-infiltrating lymphocytes (TIL) and (2) percentage of circumferential peri-tumoral (PT) area occupied by IC infiltrates. Scoring was as follows: "0": absence of IC; "1": rare, <5% of the tumor area or the PT area positive; "2": 5 to 50% of the tumor area or PT area positive; and "3": 50–100% of tumor area or PT areas positive. PD-L1 expression in a characteristic membranous pattern was recorded as (1) percentage of tumor area (tumor cells) positive and (2) percentage of IC area positive in the circumferential PT areas. The immunohistochemical staining was assessed and quantified independently by two anatomic pathologists (R.L.B. and S.G.). Any discordant results were reviewed microscopically and consensus reached.

**ctDNA**. Circulating tumor DNA was isolated and analyzed as described previously[10]. Briefly, plasma was extracted from serial patient blood samples. Bi-PAP real-time PCR assays were done using primers with dideoxynucleotide 3′ ends, specific for GNAQ and GNA11 mutations. Total human cell-free circulating DNA was quantified using a LINE1 real-time PCR assay on a serial dilution of normal DNA in each plate as standard.

**URLs**. FastQC, http://www.bioinformatics.babraham.ac.uk/projects/fastqc/; Bowtie2 2.1.0, http://bowtie-bio.sourceforge.net/bowtie2/index.shtml; Picard Tool MarkDuplicates v1.97, https://broadinstitute.github.io/picard/; HaplotypeCaller, https://software.broadinstitute.org/gatk/documentation/tooldocs/current/org_broadinstitute_gatk_tools_walkers_haplotypecaller_HaplotypeCaller.php; MuTect2, https://software.broadinstitute.org/gatk/documentation/tooldocs/current/org_broadinstitute_gatk_tools_walkers_cancer_m2_MuTect2.php; SAMtools mpileup, http://samtools.sourceforge.net/; ANNOVAR, http://annovar.openbioinformatics.org/en/latest/; cosmic80, cancer.sanger.ac.uk; GDC portal, https://portal.gdc.cancer.gov; ExAC database, http://exac.broadinstitute.org/; cbioportal, http://www.cbioportal.org/.

**Data availability**. Sequencing data have been deposited in and are available from the European Genome-phenome Archive database under number EGAS00001002761.

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

## Acknowledgements

Supported by funding from the European Commission under the Horizon 2020 program (U. M. Cure/L. Mobuchon; project number: 667787), the INCa/ITMO/AVIESAN PhD fellowship program "Formation à la recherche translationelle" (M. Rodrigues), the Cancéropôle Ile de-France, the INCa PRT-K14 UMCURE, the Institut National de la Santé et de la Recherche Médicale (INSERM), the Ligue Nationale Contre le Cancer (Labellisation), and the Institut Curie. The Institut Curie ICGex NGS platform is funded by the EQUIPEX "investissements d'avenir" program and ANR10-INBS-09-08 from the Agence Nationale de le Recherche. We thank the patients and their family members. We also acknowledge support from the Institut Curie for sample collection, banking, and processing: the Biological Resource Center and its members (O. Mariani), the Unité de Génétique Somatique and its members (G. Pierron, K. Ait-Rais), the Unité de Pharmacogénomique, the Experimental Pathology Unit and its members (D. Meseure, A. Nicolas, L. Lesage) and the Histology Laboratory, Department of Pathology and its members (A. Vincent-Salomon, M. Caly) for IHC; the next-generation sequencing team (S. Baulande, P. Legoix-Né); the Pathology Unit of the Centre Hospitalier National d'Ophtalmologie des Quinze-Vingts for sample collection and banking; the Merck Research Pathology Laboratory and its staff, Palo Alto, CA, USA (J. Yearly and staff members) for IHC performed for PD-L1; Anne-Céline Derrien and Josh Waterfall for reviewing the manuscript and providing helpful suggestions. In addition, the results here are in part based upon data generated by the TCGA Research Network: http://cancergenome.nih.gov/.

## Author contributions

M.R. and L.M. conceived the study, interpreted the data, and wrote the manuscript. A.H. and A.F. performed bioinformatics analyses. S.G. and R.L.B. performed pathological analyses. T.P. performed bioinformatics analyses. V.S. performed radiological evaluations. A.R. performed and analyzed ctDNA experiments. A.M. performed and analyzed FACS experiments. S.D. performed Sanger sequencing. V.R. performed next-generation sequencing. M.G. prepared patient specimens. M.P. performed pathological analyses. S.T. and N.C. provided patients specimens and critical advice. S.R.-R. interpreted the data and provided critical advice. F.-C.B. and O.L. interpreted the data and provided critical advice. P.M. and S.P.-N. provided patients specimens and critical advice. M.-H.S. conceived and guided the study, interpreted the data, and wrote the manuscript. All authors reviewed and approved the final manuscript.

## Additional information

**Competing interests:** M.R. and S.P.-N. received research grants from Bristol-Myers Squibb and Merck Sharp & Dohme; R.L.B. received the technical support from Merck Sharp & Dohme for IHC. The remaining authors declare no competing interests.

