## [Peer Review File · Nature Communications]

Reviewers' comments:

Reviewer #1 (Remarks to the Author):

The manuscript submitted by Rodrigues et al describes an outlier response to anti-PD1 therapy in a patient with uveal melanoma, a disease previously identified as being poorly responsive to immunological checkpoint blockade. This case underwent molecular profiling and was found to have an unusually high CpG>TpG mutation burden associated with an MBD4 germline deleterious mutation and inactivation within the tumor. Other hypermutator profiles were identified in patients carrying germline deleterious MBD4 mutations with somatic loss of heterozygosity, including one other case of uveal melanoma. Overall, this is an interesting report with potentially important clinical implications in cases harboring similar MBD4 alterations and would serve to increase awareness of these alterations. My minor comments are as follows:

1. Biopsy and analysis of the treatment refractory subcutaneous lesions would be of great interest and the findings may potentially strengthen the manuscript.
2. It would be helpful to specify the T cell populations identified (ie central and effector memory cells) on page 3, lines 53-54.
3. Page 3, Line 57: As the immune response was drug-induced, I would not consider this a "spontaneous" response.
4. Figure 2 in general is difficult to follow. In Figure 2B, the x-axis labels below each of the 6 panels are not legible. Presumably the top 3 panels of Figure 2B as well as the 3 panels of Figure 2C refer to UVM_IC, UVM_1, and GBM_4, but it would be helpful to make this absolutely clear in the legend if in fact this is the case.
5. Presumably no other cases with MBD4 mutations other than the uveal melanoma and the one GBM case of the 19 CpG>TpG hypermutated cases presented in Figure 2D... this should be clearly stated if true. It would also be of interest (eg. TDG) if other relevant alterations were identified in the MBD4 wild-type cases.
6. Overall, the manuscript could benefit from editing for grammar and clarity.

Reviewer #2 (Remarks to the Author):

This is a well presented, concise, comprehensive and intriguing case report that identifies germline mutation and subsequent somatic loss of heterozygosity of MBD4 as the cause of complete clinical response to the PD-1 inhibitor pembrolizumab in a patient with uveal melanoma. This tumour type is notoriously poor at responding to immune checkpoint inhibitors, presumably due to the extremely low mutation burden seen in most samples. In the current case, the tumour had an extremely high mutation burden, specifically characterized by C to T changes at CpG>TpG sites, which can be due to spontaneous deamination of 5-methylcytosines. The authors took a candidate gene approach to ascertain whether germline mutation of the MBD4 glycosylase might be responsible, and proved this supposition to be true. Data-mining of >10,000 TCGA tumours, including 80 uveal melanomas, identified 20 other CpG>TpG hypermutated tumours. In 2 of these (1 uveal melanoma and 1 glioblastoma) germline MBD4 mutations were found.

The importance of this work is two-fold: firstly, it provides a potential biomarker of response to PD-1

inhibition, which may be pan-cancer specific; secondly, it suggests a potential new susceptibility (or modifier) gene for uveal melanoma.

Reviewer #3 (Remarks to the Author):

The authors have identified a uveal melanoma patient who was an exceptional responder to immune checkpoint therapy. They then associate this response with a high frequency of C>T mutations in a CpG context and go on to identify a germline variant in MBD4 with LOH providing a mechanism to explain the hypermutation in this case. Clonality analysis is also consistent with this concept. Two additional tumors with a similar profile were identified, one from Curie and one from TCGA. This is an interesting report which supports the concept that MBD4 variants can be a source of hypermutation, a property of immediate clinical importance given the availability of immune checkpoint inhibitors. The authors should address the following points:

- 1) Mutation counting throughout: please clarify whether counts reflect total mutations or only non-synonymous mutations. If total, please add the count of missense mutations.
- 2) The authors identified a TCGA glioblastoma with MBD4 mutation and a CpG enriched C>T mutation signature. First, it appears that the original TCGA sample identifiers have been removed from this list which is now sorted and numbered by C>T mutation count. The original numbers should be added to the table. Interestingly the sample with the MBD4 variant is 4th on this sorted list. What was the MBD4 status of the higher ranking cases? In the main text, it would seem better to report the number (and proportion) of TCGA cases above threshold values which would be consistent with the proposed mechanism and indicate how many of these cases had an MBD4 variant. Overall, as interesting as they are, the cases with MBD4 deleterious variants seem rather rare, which begs the question of what is going on in other tumors with a similar signature.
- 3) The supplement labels one group of samples "resistant", but the term "resistant" is not used in the main text or the supplement outside of the figure. Please define and explain relevance.
- 4) The sequencing data should be submitted to a public database.

We thank the reviewers for their positive and constructive comments on our manuscript. We detail below our responses and highlight them in the manuscript. In addition, the manuscript was edited for grammar and clarity and re-organized in order to fit the *Nature Communications* standards.

Answers to Reviewer #1 comments

1. Biopsy and analysis of the treatment refractory subcutaneous lesions would be of great interest and the findings may potentially strengthen the manuscript.

We absolutely agree. Indeed, we did analyze a subcutaneous lesion progressing during pembrolizumab treatment (dark gray in figure 2a). However, the analysis of this subcutaneous tumor did not give the clue yet on its resistance to pembro. Therefore, we did not address the resistance mechanism in this manuscript.

⇒ We modified a sentence to be more clear

“To explore this outlier responder, we analyzed by whole-exome sequencing the primary tumor, the liver metastasis and a pembrolizumab-resistant subcutaneous metastasis”.

2. It would be helpful to specify the T cell populations identified (ie central and effector memory cells) on page 3, lines 53-54.

The T cell populations are now specified in the text:

- “After 2 months of pembrolizumab, proportions of blood effector memory CCR7/CD45RA⁻/CD4⁺ and CCR7/CD45RA⁺/CD8⁺ T-cells increased from 14.5% to 21.8% and from 12.2% to 48.9%, respectively”
- Figure 1d legend: “(d) Proportions of blood effector memory CCR7/CD45RA⁻/CD4⁺ and CCR7/CD45RA⁺/CD8⁺ T-cells in 12 metastatic uveal melanoma patients treated with PD-1 inhibitors, including UVM_IC (orange line).”

3. Page 3, Line 57: As the immune response was drug-induced, I would not consider this a “spontaneous” response.

The immune response in the liver metastasis was observed on a surgical sample, before any pembrolizumab infusion.

⇒ We modified a sentence to be more clear:

“These observations suggested that pembrolizumab stimulated a previously existing spontaneous cell-mediated immunity against UVM_IC tumor.”

4. Figure 2 in general is difficult to follow. In Figure 2B, the x-axis labels below each of the 6 panels are not legible. Presumably the top 3 panels of Figure 2B as well as the 3 panels of Figure 2C refer to UVM IC, UVM 1, and GBM 4, but it would be helpful to make this absolutely clear in the legend if in fact this is the case.

Figure 2 x-axis label and legend were modified:

- Instead of trinucleotides only the mutated base appears now

- "(b) Mutational patterns in the tumors of interest (above; from left to right: UVM_IC, UVM_1 and GBM_4) and in the other cases of the corresponding series (below)."
- "(c) *MBD4* mutations in germline (GI) and tumor (Tu) in the samples of interest (from left to right: UVM_IC, UVM_1 and GBM_4)."

5. Presumably no other cases with *MBD4* mutations other than the uveal melanoma and the one GBM case of the 19 CpG>TpG hypermutated cases presented in Figure 2D... this should be clearly stated if true. It would also be of interest (eg. TDG) if other relevant alterations were identified in the *MBD4* wild-type cases.

No other *MBD4* or *TDG* variant was identified in our series or in the uveal melanoma TCGA series. We did not screen the whole TCGA series for *MBD4* or *TDG* SNV, but only the top 20 hypermutated CpG>TpG cases.

- ⇒ We specified this point in the text adding the following sentences :
 - "No other sample in our UM series carried a *MBD4* or *TDG* mutation."
 - "No other *MBD4* or *TDG* mutation was identified in this series."

6. Overall, the manuscript could benefit from editing for grammar and clarity.

The revised manuscript was edited by a native English speaker (Joshua Waterfall).

Reviewer #2 (Remarks to the Author):

We thank reviewer #2 for the positive comments on our work.

Reviewer #3 (Remarks to the Author):

1) Mutation counting throughout: please clarify whether counts reflect total mutations or only non-synonymous mutations. If total, please add the count of missense mutations.

The mutation counting in this article reflects the total number of somatic variants.

- ⇒ Number of non-synonymous somatic variants has been added to the text :
 - "[...] more than 266 somatic single nucleotide variants (SNVs) per sample (more than 176 non-synonymous SNVs) [...]"
 - "[...] as well as 474 SNVs (305 non-synonymous SNVs) [...]"
 - "Of these 20 cases, Patient GBM_4 presented a glioblastoma carrying 1149 SNVs (440 non-synonymous SNVs) and a germline c.335+1G>A;p.R83Pfs*5 [...]"

2) The authors identified a TCGA glioblastoma with *MBD4* mutation and a CpG enriched C>T mutation signature. First, it appears that the original TCGA sample identifiers have been removed from this list which is now sorted and numbered by C>T mutation count. The original numbers should be added to the table. Interestingly the sample with the *MBD4* variant is 4th on this sorted list. What was the *MBD4* status of the higher ranking cases? In the main text, it would seem better to report the number (and proportion) of TCGA cases above threshold values which would be consistent with the proposed mechanism and indicate how many of these cases had an *MBD4* variant. Overall, as interesting as

they are, the cases with MBD4 deleterious variants seem rather rare, which begs the question of what is going on in other tumors with a similar signature.

1. The TCGA identifiers were removed on purpose, as it is a requirement for TCGA to re-anonymize the series when describing germline mutations.
2. Of the more than 10,000 cases from TCGA, we identified 20 hypermutated CpG>TpG cases (Page 7 line 106). Of these 20 cases, we identified germline *MBD4* mutations in 2 cases only: UVM_1 and GBM_4. We identified 3 cases with somatic, monoallelic, *MBD4* frameshift indel (2 colorectal and 1 endometrial adenocarcinomas with mismatch repair deficient). The molecular mechanism of hypermutation in the other cases remains undetermined.
3. We fully agree with reviewer #3, there may be another mutagenic mechanism explaining the other hypermutated CpG>TpG cases that remains to be discovered.

⇒ We modified 3 sentences to be more clear:

“We further analyzed the pan-cancer TCGA series (>10,000 tumors; Supplementary Table 1) and identified 4,831 hypermutated tumors (>200 SNVs per tumor) of which 20 cases, including UVM_1, were enriched in CpG>TpG mutations ((CpG>TpG)/SNVs>0.6; Fig. 2d).”

“The three other hypermutated CpG>TpG glioblastoma cases did not present identifiable deleterious *MBD4* or *TDG* mutation.”

“To be noticed, three of these 20 hypermutated cases carried somatic *MBD4* indels together with mismatch repair deficiency (two colorectal and one endometrial adenocarcinomas); the molecular mechanism of hypermutation in the other cases remains undetermined.”

3) The supplement labels one group of samples “resistant”, but the term “resistant” is not used in the main text or the supplement outside of the figure. Please define and explain relevance.

The “resistant” sample is a subcutaneous lesion progressing under pembrolizumab (figure 1c and page 3 line 49).

⇒ We modified a sentence to be more clear.

“To explore this outlier response, we performed whole-exome sequencing of the primary tumor, the liver metastasis, and a pembrolizumab-resistant subcutaneous metastasis, as well as constitutional DNA”

4) The sequencing data should be submitted to a public database.

A “data availability” section was added: “Sequencing data have been deposited in and are available from the European Genome-phenome Archive database under number EGAS00001002761”

REVIEWERS' COMMENTS:

Reviewer #1 (Remarks to the Author):

The authors have adequately responded to the comments of the previous reviewers. The manuscript which reports that MBD4 loss results in high CpG>TpG mutational burden and may explain outlier responses to immunological checkpoint blockade is novel and of interest to the cancer research community. I have no further comments/suggestions.

Reviewer #3 (Remarks to the Author):

The authors have responded adequately to the previous critique.

Reviewer #1 (Remarks to the Author):

The authors have adequately responded to the comments of the previous reviewers. The manuscript which reports that MBD4 loss results in high CpG>TpG mutational burden and may explain outlier responses to immunological checkpoint blockade is novel and of interest to the cancer research community. I have no further comments/suggestions.

Reviewer #3 (Remarks to the Author):

The authors have responded adequately to the previous critique.

We thank the reviewers for their positive comments on our revised manuscript.